# Access to Dental Care and Depressive Illness: Results from the Korea National Health Nutrition Examination Survey

**DOI:** 10.3390/medicina56040189

**Published:** 2020-04-19

**Authors:** Kyung Hee Choi, Sangyoon Shin, Euni Lee, Seok-Woo Lee

**Affiliations:** 1College of Pharmacy, Sunchon National University, Suncheon, Jeollanam-do 57922, Korea; 2College of Pharmacy & Research Institute of Pharmaceutical Sciences, Seoul National University 1 Gwanak-ro, Gwanak-gu, Seoul 08826, Korea; 3Departments of Dental Education and Periodontology, School of Dentistry, Dental Science Research Institute, Chonnam National University, Gwangju 61184, Korea

**Keywords:** depression, oral health, dental visit, real-world data

## Abstract

*Background and Objectives*: Recent evidence suggests that oral health is associated with various systemic diseases including psychiatric illnesses. This study examined the association between depression and access to dental care in Korean adults. *Materials and Methods*: A cross-sectional evaluation was performed using data from the Sixth Korea National Health and Nutrition Examination Survey 2014. The general characteristics of the participants, the current depression status, and issues with access to dental care were collected to evaluate the factors for not being able to make dental visits according to care needs. *Results*: The study population comprised a total of 5976 participants who were 19 years of age and older and represented 40.7 million Koreans. A multivariable logistic regression analysis with weighted observations revealed that participants with current depressive illness were about two times more likely to express that they could not make dental visits in spite of their perceived care needs (adjusted odds ratio (OR) = 2.097; 95% confidence interval (CI) 1.046–4.203). The reasons for not making dental visits included financial problems, perceived importance of the dental problem, and fear of visiting dental professionals. *Conclusions*: Korean adults with current depressive illness were less likely to make dental visits when they had dental care needs. To improve dental health accessibility for patients with depressive illness, coordinated efforts can be considered involving multidisciplinary health care professionals.

## 1. Introduction

As the World Health Organization described, oral health is a key indicator of overall health, well-being and quality of life [1], encompassing a range of oral and other systemic conditions. In recent years, evidence has suggested the relationship between periodontal disease and other chronic systemic diseases, including respiratory disease, cardiovascular disease, diabetes, adverse pregnancy outcomes, and even various types of cancers [2,3,4,5]. Recent studies strongly indicate that inflammatory and immunological responses triggered by oral bacterial biofilms may damage distant tissues or organ systems [6,7]. For these reasons, poor oral health can be a serious threat to overall health, and the importance of routine dental care is emphasized [8,9].

People with mental illness have also shown a higher rate of physical health problems, including cardiovascular disease, chronic lung disease, diabetes, and cancer, compared to those without these disorders [10,11,12]. Many studies have demonstrated that patients with mental illness have a twofold increase in mortality risk [13,14]. Compared to the general population, patients with mental illness have been reported to have higher instability in their employment status, lower income, and lower social status [12,15], and they have difficulties in accessing health care providers due to the economic burden and dissatisfaction with their services [10,12,16]. Patients who are taking medications for mental illnesses may also experience xerostomia, which can lead to poor oral health [17]. Patients with mental illness are also more likely to exhibit adverse lifestyle behaviors, including cigarette smoking, alcohol consumption, or behavioral problems, which can lead to ignoring oral health care [4,12,18,19].

While the relationship between oral and systemic health has been extensively documented in recent years [6,7], scientific evidence addressing the association between oral health and patients’ social and psychological status is relatively scarce. Recent literature suggests that chronic periodontitis is associated with mental disorders [20,21,22,23], as it was postulated that inflammatory mediators resulting from the development of chronic periodontitis may affect the central nervous system [24,25].

Social variables, including social class, education, employment status, personal income, urbanization, and vulnerability have been evaluated [26,27,28]. Published studies have documented that negative mental status, such as anxiety or depression, is associated with a tendency towards poor oral health including the presence of chronic periodontitis, but the majority of these studies were performed using a small sample size or were limited to a special population [3,29,30,31]. Therefore, it is important to precisely assess the relationship between impaired mental status and the condition of oral and periodontal health. The aim of the present study was to evaluate a null hypothesis that depression, as a mental illness, is not an associated factor against access to dental care among Korean adults using national-level real-world data.

## 2. Materials and Methods

### 2.1. Data Source and Study Population

This cross-sectional study used data of the Sixth Korea National Health and Nutrition Examination Survey (KNHANES VI-1) that was conducted by the Korea Centers for Disease Control and Prevention (KCDC) [32]. KNHANES is a nationwide, population-based, cross-sectional survey that examines the general health and nutritional status of Koreans, and it has been conducted by the Korean Ministry of Health and Welfare and KCDC since 1998 [32]. It comprises the following three surveys: a health examination survey, a health interview survey, and a nutritional survey. The KNHANES 2014 survey used a complex, stratified, multistage probability-cluster sampling design representing non-institutionalized civilians in South Korea [32].

The study included a total of 5976 survey participants who were 19 years and older. They were an unweighted study sample in the year 2014 with a mean (SD) age of 51.42 years (SD 18.29). Data of the sociodemographic characteristics (i.e., age, gender, education, marital status, and insurance status), current depression status, and access to a dental office with perceived dental problems were collected. The KNHANES data allowed researchers to obtain national-level estimates using the assigned sampling weight variable when they were aggregated to generate weighted frequencies [32].

### 2.2. Definitions of “Current Depression” and “No-Dental Visit with Care Needs”

Data of “current depression” status were collected from the survey participants using a variable from the health interview survey. The interviewer asked if the participants “*currently suffer from depression*” among those who have ever been diagnosed with depression [33]. A “*Yes*” response was considered as having “current depression” status [34]. The study used a variable named *BM12* provided by the KNHANES to describe “no-dental visit with care needs” by determining if a participant indicated “*Yes*” to the following survey question: “*In the past one year, have you ever thought of the need for dental care but could not make a dental visit?*” in the health examination survey under the oral health section [34]. For those who answered “*Yes*”, a follow-up question was given as *“What was the major reason for not being able to make a dental visit?”* Possible answers for the reasons included *“financial reasons”*, *“distance to the dental office”*, *“cannot miss work or school”, “physical difficulties or health issues”, “no childcare”*, *“felt it was a less important problem”*, *“afraid of dental visit”*, and *“others.”* For the reasons marked as *“others,”* verbatim text descriptions were collected.

### 2.3. Statistical Analysis

The analysis of the study focused on estimating the prevalence of “no-dental visit with care needs”, evaluating its associated factors, and describing the major reasons for not being able to make a dental visit in spite of the care needs of participants with current depression status. For obtaining national-level estimates, i.e., weighted frequencies, assigned sampling weights were aggregated to account for the complex sampling design [32]. Descriptive statistics were used to calculate the frequencies of sociodemographic characteristics and the prevalence of depression as well as access to the dental office with perceived care needs.

Univariable and multivariable logistic regression analyses were performed using weighted data with valid data points excluding missing data. To assess the factors associated with “no-dental visit with care needs” as the dependent variable, a multivariable logistic regression model was constructed with the current depression condition (yes/no) as the independent variable and the adjusted odds ratios (ORs) with 95% confidence intervals (CIs) was determined after adjusting for confounders, such as age, gender, level of education, marital status, and health insurance type reflecting the Korean health insurance structure. The estimation of the national-level frequencies and logistic regression analyses were carried out using weighted data, and all statistical analyses were conducted using SPSS version 23.0 (SPSS Inc., Chicago, IL).

We analyzed the major reasons for not making a dental visit with perceived care needs using descriptive statistics presenting the proportion of specific reasons with a subgroup analysis. The subgroup analysis included a total of 90 participants using unweighted data since a reliable estimation of the weighted frequencies was not feasible. The significance level was set at *p* < 0.05. The study used publicly-available national-level data without patient identifiers or linkable information. The study was approved by the Institutional Review Board of the Chonnam National University Dental Hospital, Gwangju, Korea (Approval number: CNUDH-EXP-2018-005, approved on 20 November 2018).

## 3. Results

A total of 7550 participants completed the KNHANES VI 2014, and 79.2% (5976 participants) of the participants were adults aged 19 years of age and older (Figure 1). Descriptive characteristics of the study population representing 40.7 million Koreans are summarized in Table 1. The mean age of the study population was 46.3 years (95% CI 45.4–47.2), and the gender distribution was similar between males and females (49.4% and 50.6%, respectively, Table 1). The majority of the participants were married (78.0%), had graduated from high school or higher (74.2%), and were covered by health insurance through their workplace (63.8%). The prevalence of current depression status was 2.6%. In addition, 29.2% of the study participants reported that they could not visit the dental office in the last year with perceived needs for dental care (Table 1).

Univariate and multivariate associations with “no-dental visit with care needs” were evaluated by a logistic regression analysis using weighted data. Age, level of education, marital status, and type of health insurance were not significantly associated with access to dental care (Table 2). Although the univariate analysis showed that male participants and those who were 65 years or older were more likely to make visits to dentists with dental care needs than their counterparts, the associations disappeared in the multivariate model. A significant association was found between current depression status and “no-dental visit with care needs” (adjusted ORadj = 2.262, 95% CI 1.138–4.497), indicating that participants with current depression condition were about two times more likely to express their access problems to dentists in spite of their perceived care needs (Table 2).

From the total population, 90 participants reported they did not make a dental visit in spite of their needs for dental care, and the reasons for not making dental visits are displayed using unweighted frequency data by current depression status (Figure 2). The most common reasons for not making the dental visit were similar. The reasons described by participants with vs. without depression included financial problems (49.2% vs. 48.3%), perceived importance of the dental problems (21.3% vs. 17.2%), and fear of making dental visits (14.8% vs. 17.2%), respectively (Figure 2). In addition, participants with depression added geographic location (i.e., long distance to the dentist) or resource availability (i.e., no childcare) as the reasons for their access problems. Others included “hospitalization to a psychiatry hospital” for a patient with current depression and “pregnancy” from a survey participant without a current depression condition.

## 4. Discussion

One of the strengths of this study was the value of the source data. The KNHANES VI data provided a national-level estimation and allowed to construct an analytic model using weighted data. Our findings indicated that depression was one of the barriers to dental care in the Korean population. Since real-world data are considered an important source to guide future research designs and to generate real-world evidence that can be used in the decision-making processes for healthcare [35,36], we believe that our findings derived from national-level real-world data could provide important insights into missed opportunities in dental care access for patients with depressive illnesses.

The findings of our study indicated that participants with current depression were more likely to experience access problems to dental services than their counterparts. Our findings were consistent with those reported by Antilla et al., which showed that patients with depressive symptoms made less frequent dental visits than subjects with no depressive symptoms or only a few depressive symptoms [37]. Additional studies have also described depression as a risk factor for poor oral health. Skośkiewicz-Malinowska et al. reported that people aged 65 years and over exhibited more severe depression along with a higher number of missing teeth, number of decayed teeth, and prevalence of xerostomia [30]. In another study, depression was significantly associated with the number of decayed teeth only in participants aged 35–54 years and not in the other age groups [38]. A high susceptibility to depression was also found to not play a significant role in the etiology and severity of periodontitis [39]. However, one case control study found that there was a direct correlation between the severity of periodontal disease and the severity of depression [40]. Collectively, it appears that a positive association exists between depression and dental caries, tooth loss, and edentulism in adults and elderly people [41]; however, studies documenting the relationship between depression and patient-reported access barriers to dental care are lacking.

Poorer oral health in subjects with depression has been explained by neglected oral hygiene practices, avoidance of optimal dental care, and adverse drug effects caused by antidepressants [42,43], leading to an increased risk for the development of dental caries and chronic periodontitis. Regular dental visits are essential to maintain optimal oral health [44] in order to decrease the inflammatory burden that might damage oral and periodontal tissues. Since immuno-inflammatory mediators resulting from these inflammatory responses may get access to central nervous system, it is crucial to decrease the inflammatory burden in order to decrease potential mental disorders [24,25].

The reasons for not seeking regular dental care may include negative memories from previous dental visits, economic deterioration, and unwillingness to improve oral health [11,12,16,45]. The findings of the current study showed that the common reasons for not making dental visits by depression status were similar with “*financial problem*” followed by “*less important problems*”. Although it is plausible that the oral health problem was affected more by the severity of the depression status [30,46], our study was limited in making inferences on the exact reasons for why participants with or without depression did not make dental visits. For example, most people with psychiatric disorders have a fear of dental visits due to the characteristics of the mental disease itself [16,47]. Therefore, further studies including more number of participants are required to elucidate the underlying causes of access barriers to oral health care in people with depression.

We believe that developing a collaborative network among health care providers between hospital and community settings can be a potential strategy to improve patient-centered dental care for patients with mental illness, including depressive disorders. A study by Corridore et al. reported that patients with psychiatric illness often lacked motivation for routine dental care due to cigarette smoking, alcohol consumption, and lower frequency of toothbrush use [4]. In a recent study, patients who accepted treatment for depressive disorders were observed to be five times more likely to fear dental treatment compared to control subjects, and more than half of these patients had cancelled or failed to keep dental appointments due to fear [48]. Kisely et al. suggested that health care providers should cooperate with oral health management by developing oral health assessment tools to facilitate the optimal dental care [49]. In one qualitative study, primary care was emphasized as a core institution that could improve oral hygiene with easier accessibility for the patients and timely intervention to manage depressive symptoms [18]. It was predicted that one in every five patients who visited a dentist experienced clinically-significant symptoms of depression [50]. Therefore, dentists are well positioned to make timely referrals to specialists or patients’ primary care physicians and could play an active role in a multidisciplinary care network involving medical professionals, allied health professionals, and pharmacies [51] in the community. In summary, the importance of teamwork between clinicians, psychiatrists, dentists, and other healthcare professionals cannot be overemphasized.

This study has several limitations. First, due to the nature of the KNHANES as a cross-sectional study, the causality between current depression and access to dental care could not be established. Second, the exact reasons for not making a dental visit could not be generalized due to the small number of participants. In addition, our study was limited in making inferences on the exact reasons for why participants with or without depression did not make dental visits. Third, our evaluation was focused on the perceived dental needs as provided by the KNHANES data. As indicated in the literature, perceived dental needs are different from objective dental needs [52]. Therefore, interpretations of our findings should be limited to reflect only subjective needs by the participants. Lastly, findings from the KNHANES data on health care access may provide limited applicability to other countries and societies with different health care systems. In spite of these limitations, we believe that the KNHANES data can be an informative tracking tool for chronic diseases and an effective research tool for health service researchers by generating real-world evidence of dental health and its association with other clinical conditions or health-related behaviors.

## 5. Conclusions

The results of the present study revealed that Korean adults with current depressive illnesses were less likely to make dental visits corresponding to their dental care needs. Since the relationship between the severity of depression and detailed needs of their dental care could not be determined from the survey, well-designed prospective studies may be necessary to determine the clinical implications of our findings. To improve access to dental care for people with depressive illnesses, coordinated efforts can be considered involving multidisciplinary health care professionals.

## Figures and Tables

**Figure 1 medicina-56-00189-f001:**
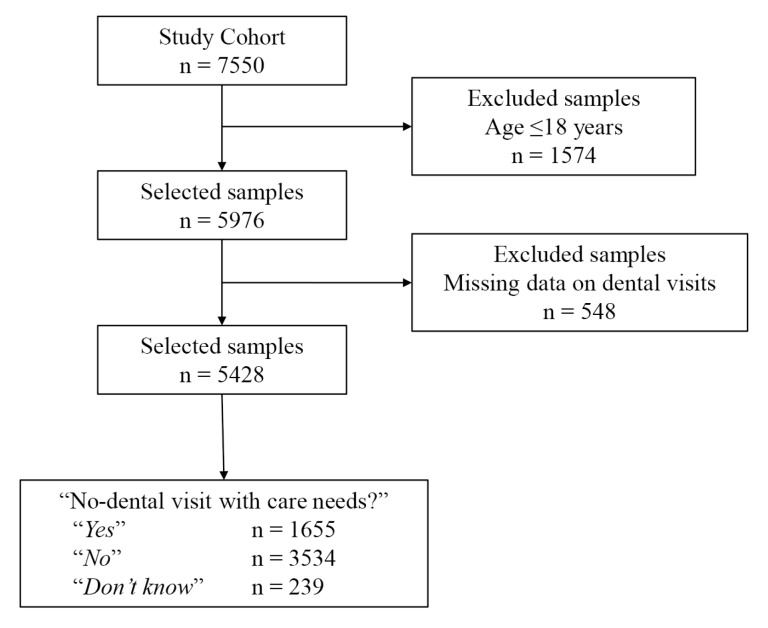
Flow chart for sample selection process.

**Figure 2 medicina-56-00189-f002:**
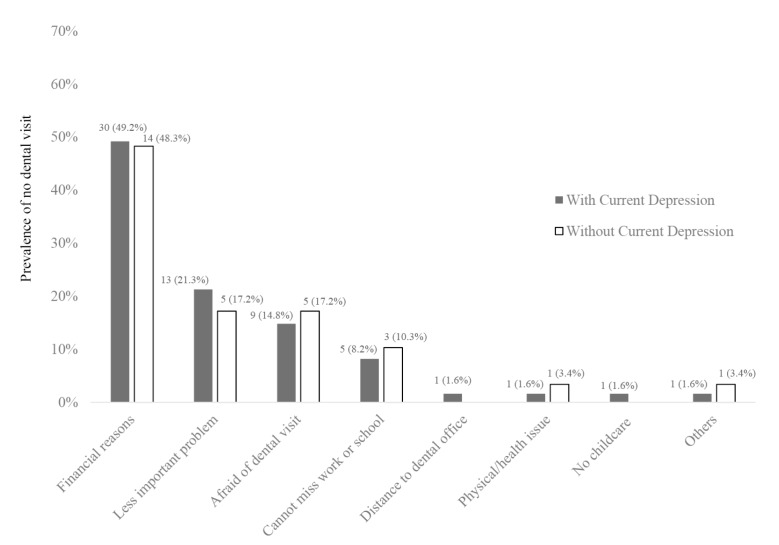
Major reasons not making a dental visit with perceived care needs by depression status using unweighted data (*n* = 90).

**Table 1 medicina-56-00189-t001:** Descriptive characteristics of the study population from 2014 Korea National Health and Nutrition Examination Survey (KNHANES).

Characteristics		Unweighted n (%)	Weighted n (%) *
Total participants		5976 (100.0)	40,767,232 (100.0)
Age (years)	19–34	1186 (19.8)	11,530,548 (28.3)
	35–49	1571 (26.3)	12,343,891(30.3)
	50–64	1633 (27.3)	10,507,234 (25.8)
	65 and over	1586 (26.5)	6,385,559 (15.7)
Gender	Female	3292 (58.0)	20,625,518 (50.6)
	Male	2385 (42.0)	20,141,714 (49.4)
Education	Middle school or less	1757 (35.2)	9,188,642 (25.8)
	High school graduate	1644 (32.9)	13,614,618 (38.3)
	College graduate or higher	1589 (31.9)	12,755,125 (35.9)
Marital status	Single	902 (15.1)	8,834,862 (22.0)
	Married	5073 (84.9)	31,914,132 (78.0)
Health Insurance	Self-employed	1834 (32.9)	13,196,940 (33.0)
	Workplace	3532 (63.3)	25,537,110 (63.8)
	Others ^§^	210 (3.8)	1,313,596 (3.3)
No-dental visit with care needs ^†^	Yes	1655 (27.7)	11,907,286 (29.2)

* Weighted frequencies were calculated using the sampling weight variable accounting for the complex sampling design provided by the Korea Centers for Disease Control and Prevention.^§^ ‘Others’ group included Medical Aid class 1 and 2, no health insurance, or unknown. ^†^ “No-dental visit with care needs” from the KNHANES survey described that the participants perceived the needs for dental care but could not visit the dental office in the last 1 year.

**Table 2 medicina-56-00189-t002:** Associated factors against access to dental care with perceived dental problems from 2014 Korea National Health and Nutrition Examination Survey (KNHANES).

Participant Characteristics	Unadjusted OR (95% CI) *	Adjusted OR (95% CI) *
Current depression ^†^		
No	1	1
Yes	2.033 (1.027–4.027) ^§^	2.262 (1.138–4.497) ^§^
Age (years)		
19–34	1	1
35–49	0.996 (0.818–1.214)	0.587 (0.184–1.875)
50–64	0.905 (0.757–1.080)	1.705 (0.536–5.426)
65 and over	0.648 (0.536–0.784)^§^	0.827 (0.212–3.232)
Gender		
Female	1	1
Male	0.817 (0.710–0.941)^§^	0.424 (0.167–1.078)
Education		
Elementary school	1	1
Middle school	1.098(0.861–1.400)	0.355 (0.124–1.015)
High school	1.098 (0.919–1.312)	1.081 (0.453–2.580)
College graduate or higher	1.033 (0.862–1.240)	0.589 (0.199–1.744)
Marital status		
Single	1	1
Married	0.947 (0.779–1.153)	0.490 (0.139–1.736)
Health Insurance		
Workplace (employment)	1	1
Self-employed (community)	1.126 (0.846–1.499)	1.053 (0.427–2.601)
Others **	1.048 (0.906–1.213)	1.866 (0.884–3.941)

* Odds Ratios (ORs) of likelihood of “*no-dental visit with care needs*” were based on weighted data. Adjusted OR refers to OR adjusted by all covariates included in the table. ^†^ “Current depression” status was indicated in the KNHANES survey as currently having depression among those who were diagnosed with depression. ^§^
*p* < 0.05, ** ‘Others’ included Medical Aid class 1 and 2, no health insurance, or unknown.

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
