# Peer review of "Access to Dental Care and Depressive Illness: Results from the Korea National Health Nutrition Examination Survey"

_medicina, 2020, doi:10.3390/medicina56040189_

Round 1

Reviewer 1 Report

Overall this is an interesting paper with important policy and social implications for improving access to dental care for those suffering depression.  There are however a few points in the paper which require clarification and a more cautious approach interpreting the data.

  1. In the abstract and elsewhere the authors refer to "patients" rather than survey "participants".  This also applies presumably to the sub-sample who claim they "currently suffer from depression".  There is only the self-reported evidence at interview which appears to relate to this being a clinical diagnosis which may lend itself to the term "patients" for this sub-sample.  This needs to be clarified.
  2. The question related to identifying those with a perceived need for dental care AND who could not make a dental visit - are two separate concepts - therefore more caution is required in the discussion when the reasons for those with depression are compared with those who are not depressed.
  3. The authors note that the data is cross-sectional and cannot therefore be used to "predict" outcomes, yet the text refers to the "predictive factors for "no-dental visit with care needs".  Consistency in ensuring associations are not assumed predictive should be clarified in the text.
  4. Perceive dental needs are distinct from objective dental needs - often under estimating objective needs.  Perhaps a reference to the report by Farmer et al (2017). Comparing self-reported and clinically diagnosed unmet dental treatment needs using a nationally representative sample. J Public Health Dent 77: 295-301 would be a useful reference.
  5. There are some minor editorial issues with the References - for example the first reference (1) seem incomplete and reference 25 does not need the associative agencies.

Reviewer 2 Report

The aim of this study was to examined the assocation between depression and access to dental care in Korean adults with a large sample. Globally, poor oral health of individuals with mental disorders is a major public health issue. This article is timely and generally well-written paper and the design of the study is very interseting. However, there are a number of questions that arise in the reading of this paper that dampens my enthuisasm and that are reported below.

Major concern

-To describe the reasons for not making a dental visit with perceived care needs, the study was conducted with only 90 participants. I understand the inability using unweighted data since a reliable estimation of the weighted frequencies was not feasible. But, why 90 participants and not more? and how the participants were recruting?

-A suggestion in the title, the authors could introduced "a feasibility study"

Minor

Table 2: Predictive factors against access to dental care with perceived dental problems, From 2014 Korea Naional Health and Nutrition Examination Survey (KNHANES)? 

Round 2

Reviewer 2 Report

The authors responded all of my comments

Author Response

The reviewer stated that "The authors responded all of my comments". in a previous report.